

# Non-linear effects of pore pressure increase on seismic event generation in multi-degree-of-freedom rate-and-state model of tectonic fault sliding.

Sergey B. Turuntaev[1,2,3], Vasily Y. Riga[2]

[1]Institute of Geosphere Dynamics, Russian Academy of Sciences, Moscow, 119334, Russian Federation (s.turuntaev@gmail.com),

[2]Moscow Institute of Physics and Technology, Moscow, 141701, Russian Federation,

[3]All-Russian Research Institute of Automatics, Moscow, 127055, Russian Federation

*Correspondence to*: Sergey B. Turuntaev (s.turuntaev@gmail.com)

**Abstract.** The influence of fluid injection on tectonic fault sliding and generation of seismic events was studied by multi-degree-of-freedom rate-and-state friction model with two-parametric friction law. A system of blocks (up to 25 blocks) elastically connected with each other and connected by elastic springs to a constant-velocity moving driver was considered. Variation of the pore pressure due to fluid injection led to variation of effective stress between the first block and the

substrate. Initially the block system was in steady-sliding state, then its state was changed by the pore pressure increase. The influence of the model parameters (number of the blocks, the spring stiffness, velocity weakening parameter) on the seismicity variations were considered. Various slip patterns were obtained and analysed.

## 1 Introduction

Despite the fact that the rate-and-state model of friction was proposed in the second half of the previous century, the interest

to it has increased in recent years. The rate-and-state model (Gu et al., 1984, Dieterich, 1992, Abe and Kato, 2013) was adopted as a quite appropriate basis for describing seismic processes in the Earth crust and for modelling relevant geophysical systems. Currently, it is believed that this model describes the seismic process most adequately.

Brace and Byerlee (Brace and Byerlee, 1966) proposed to consider unstable frictional sliding along tectonic faults as a model of earthquakes. The model included a suggestion that a cohesion existing in some parts of tectonic fault prevents free

slipping along it and leads to an accumulation of a shear stress to a critical level, after which the slip and the earthquake occur.





Peculiarities of the friction force dependence on the duration of the stationary state of the contact and on the velocity of the motion along the fault was examined by Dieterich (Dieterich, 1992). Gu et al. (Gu et al., 1984) experimentally investigated various modes of the frictional movements and determined empirical constants which values are used in many modern variants of the rate-and-state equation.

The rate-and-state equation was considered by Hobbs (Hobbs, 1990) by means of nonlinear dynamics methods. Change of the friction was studied as a function of displacement and velocity at a variation of the stiffness coefficient in the rate-and-state equation. A similar approach was implemented by Erickson et al. (Erickson et al., 2008); they examined an appearance of chaotic solutions in the one-parameter velocity-dependent friction equation.

Abe and Kato (Abe and Kato, 2013, Abe and Kato, 2014) examined two- and three-degree-of-freedom spring-block models
with one-parameter rate-and-state friction law and obtained different slip patterns for such system. By varying stiffness parameters, they obtain periodic recurrence of seismic and aseismic events and several types of seismicity chaotic behaviour. Turuntaev et al. (Turuntaev et al., 2012) showed that the man-made impact on the underground leads to an increase in the "regularity" of the seismic regime. To explain the increase in the seismic regime regularity, a model of fault motion defined by the two-parameter velocity dependent friction law was considered.

In the presented paper, we consider a two-parameter type of the friction law in multi-degree-of-freedom spring-block model and change the value of critical shear stress in the rate-and-state equation in suggestion that this is the value varied by human impact (by fluid injection and corresponding pore pressure change). Here we use classical pore-elastic model of radial filtration of injected fluid to calculate the typical pore pressure change.

## 2 The model description

### 2.1 Spring-block model

The tectonic fault model proposed by Burridge and Knopov (Burridge and Knopov, 1966) looks like a system of blocks elastically connected with each other (Fig. 1 a, b). Each block moves under net action of elastic forces from adjacent blocks and driver and friction force from the stationary substrate. Here, the multi-degree-of-freedom system is investigated. Every block of mass $m_i$ is connected by a spring of stiffness $k_l$ to the driver moving at a rate $v_{pl}$, and linked with each other by
springs of stiffness $k_{n-1,n}$. The motion equation may be written as Eq. (1):

$$\begin{cases} m_1\ddot{x}_1 = k_1\big(v_{pl}t - x_1\big) - k_{12}(x_1 - x_2) - F_{fr1} \\ m_2\ddot{x}_2 = k_2\big(v_{pl}t - x_2\big) + k_{12}(x_1 - x_2) - k_{23}(x_2 - x_3) - F_{fr2} \\ \qquad\qquad\qquad ... \end{cases} \qquad (1)$$

where $F_{fri}$ is the force of friction between the block number $i$ and the substrate, $t$ is time and $x_i$ is the displacements of the blocks relatively to the driver.




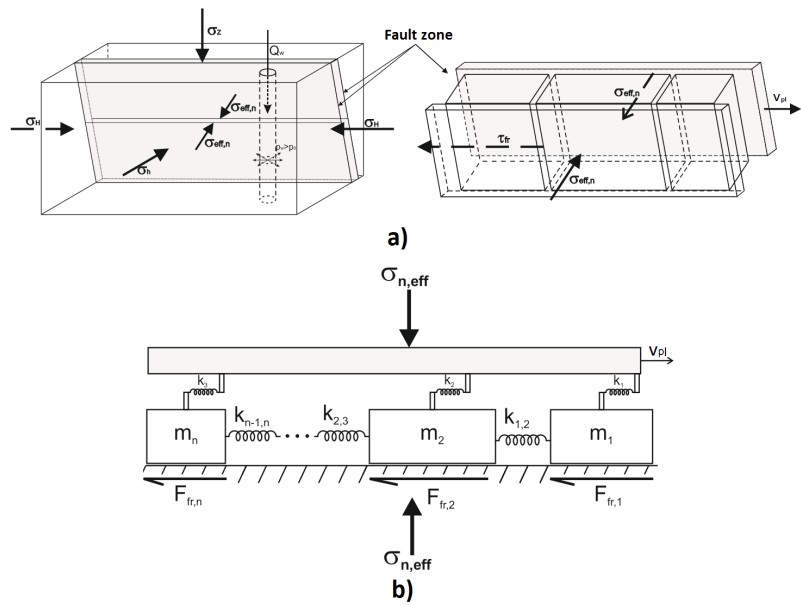

Figure 1: a) the block model of active tectonic fault; b) schematic diagram of a multi-degree-of-freedom spring-block model.

We assume that the friction shear stress at block boundary obeys the following two-parameter friction law:

$$\tau = \tau^* + A\,ln\left(\frac{v}{v^*}\right) + \theta_1 + \theta_2 \qquad (2)$$

$$\tau^* = \tau_0 + \mu(\sigma_n - p) \qquad (3)$$

$$\dot{\theta}_i = -\frac{v}{L_i}\left[\theta_i + B_i ln(v/v^*)\right] \qquad (4)$$

where $\theta_1$, $\theta_2$ are the state parameters, $A$ and $B_1$, $B_2$ are constants that represent the rate and time dependences of the friction, respectively, $L_1, L_2$ are characteristic slip distances, $v^*$ is a reference velocity, $\sigma_n$ is a normal stress, $p$ is fluid pore pressure, $\tau_0$ is a cohesion, $\mu$ is Coulomb friction coefficient, $\tau^*$ is a critical stress. Here, the values of constants $A$, $B_i$, $L_i$ were taken from the experiments of Gu et al. (Gu et al., 1984).

10    As it was shown by Gu et al, 1984, that if $A-B_1-B_2<0$ the friction shows velocity weakening, which can lead to stick-slip motion, otherwise, if $A-B_1-B_2\geq0$ the friction shows velocity strengthening. For single-degree-of-freedom spring-block model with the spring stiffness $k$, the so-called critical stiffness $k_{cr}$ (per unit area of block surface) is defined by the Eq. (5):





$$k_{cr} = \frac{2A}{L_1 + L_2}\left[(\beta_1 - 1) + \rho^2(\beta_2 - 1) + 2\rho(\beta_1 + \beta_1 - 1)\right.$$
$$\left. + \sqrt{\{[(\beta_1 - 1) + \rho^2(\beta_2 - 1)]^2 + 4\rho^2(\beta_1 + \beta_1 - 1)\}}\right]/(4\rho)$$

(5)

where $\beta_1 = \frac{B_1}{A}$, $\beta_2 = \frac{B_2}{A}$, $\rho = \frac{L_1}{L_2}$

If $k < k_{cr}$ and $A - B_1 - B_2 < 0$ the stick-slip occurs. Let us suppose that all the blocks have the same friction parameters and stiffness, and that these parameters satisfy the conditions for stick-slip. Initially, all blocks are moving with the velocities

equal to the driver velocity. To study the difference between the injection induced seismicity and the natural seismicity, two sets of numerical calculations were conducted (Set 1 and Set 2). In the first set ("natural" seismicity case), a perturbation in a form of an instant increase of the first block velocity was introduced equal to the velocity of the driver (as in Hobs, 1990). In the second set, the pore pressure in the boundary between the first block and the substrate was increased with time in accordance with pore-elastic equation solution.

The parameters in all the simulations were the followings: $B_1 = 3.3\cdot10^4$ Pa, $B_2 = 2.772\cdot10^4$ Pa, $L_1 = 2.5\cdot10^{-7}$m, $L_2 = 5.2\cdot10^{-6}$ m, $v_{pl} = 10^{-9}$ m/s (3.2 cm/year), $k_s = 9.04 \cdot 10^9$ Pa/m (stiffness per unit area of block), $\tau^* = 99\,MPa$; mass obeyed the condition $\frac{mv_{pl}^2}{AS} \ll 1$, $S$ was the area of the block contact with the substrate. By using such a small mass, we can neglect inertness of the system and Eq. (5) will be relevant for our system. For both sets of the calculations, two cases were considered, which differed by the values of $A$ and corresponded $k_{cr}$ (Table 1). It was shown (Gu et al., 1984, Hobbs, 1990),

that the one block system will move chaotically in Case 1 and periodically in Case 2.

**Table 1**

| Case | $A$ | $k_{cr}$ |
|------|-----|----------|
| 1 | $3.3\cdot10^4$ Pa | $1.06\cdot10^{10}$ Pa/m |
| 2 | $3.2\cdot10^4$ Pa | $1.11\cdot10^{10}$ Pa/m |

### 2.2 Pore pressure change

To estimate the pore pressure change, we considered radial flow of fluid in an infinite homogeneous reservoir of constant

thickness from the injection well with a negligibly small radius (Fig. 2). The initial reservoir pressure was assumed to be the same everywhere and equal to $p_0$. Volumetric flow rate of the well was constant and equal to $Q_0$. The assumptions were follows: porosity and permeability were constant (independent of the pressure), fluid had small and constant compressibility. To express the condition for constant flow rate $u_r$ at the wellbore, the Darcy's law was used:


$$u_r = -\frac{k}{\mu}\frac{dp}{dr}$$

$$Q_0 = 2\pi h r \cdot u_r = -\frac{2\pi h k}{\mu} r \frac{dp}{dr} \tag{6}$$

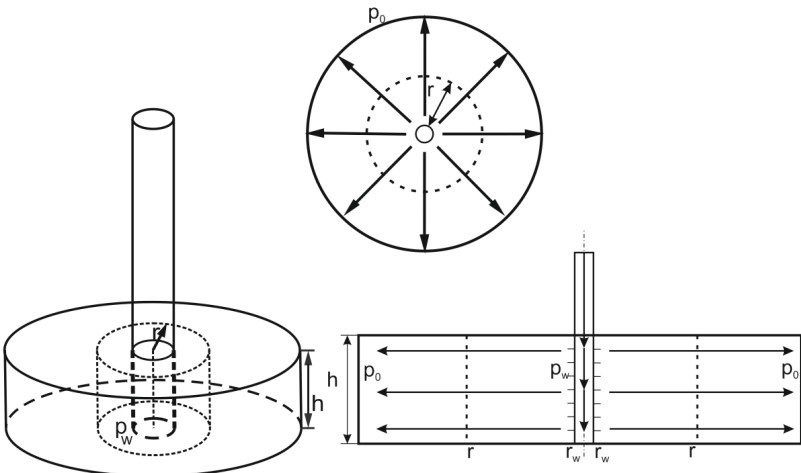

**Figure 2: Radial flow in homogeneous reservoir.**

So we got standard diffusivity equation, where $D = \frac{k}{\varphi \mu c}$ is the hydraulic diffusivity (Matthews et al., 1967):

$$\begin{cases} \dfrac{\partial p(r,t)}{\partial t} = D\left(\dfrac{\partial^2 p(r,t)}{\partial r^2} + \dfrac{1}{r}\dfrac{\partial p(r,t)}{\partial r}\right) \\[2mm] Q_0 = -\dfrac{2\pi h k}{\mu} r \dfrac{dp}{dr}\Big|_{r=r_w}, (r_w \to 0) \\[2mm] p(+\infty, t) = p_0 \\[1mm] p(r,0) = p_0 \end{cases} \tag{7}$$

5   The solution of this equation with the above initial and boundary conditions reads:

$$p = \frac{Q_0 \mu}{4\pi k h} Ei(\frac{r^2}{4Dt}) + p_0 \tag{8}$$

$$Ei(t) = \int_x^\infty \frac{e^{-t}}{t} dt \tag{9}$$

The values of parameters used in the calculations were close to the parameters of Basel project (Häring et al., 2008, Dinske, 2010): $r = 100$ m, $Q_0 = 1.5$ m³/min, $p_0 = 44$ MPa, $\mu = 0.284$ Pa·s, $h = 46$ m, $k = 4$ mD. We stopped the pressure growth




at the first block boundary when it exceeded the value 64 MPa (corresponding time is approximately $7.13 \cdot 10^6$ s, Fig. 3).

Instead of exponential integral $Ei$ (9) we used its approximation (Abramovitz and Stigan, 1979):

$$Ei(x) = \begin{cases} -ln(\gamma_1 x) & , & 0 < x \le 0,01 \\ -ln(\gamma_1 x) + a_1 x + a_2 x^2 + a_3 x^3 + a_4 x^4 + a_5 x^5, & & 0,01 < x \le 1 \\ \left(\dfrac{x^2 + b_1 x + b_2}{x^2 + c_1 x + c_2}\right)\dfrac{e^{-x}}{x} & , & 1 < x < +\infty \end{cases} \tag{10}$$

where $a_1 = 0.99999193$; $a_2 = -0.24991055$; $a_3 = 0.05519968$; $a_4 = -0.00976004$; $a_5 = 0.00107857$; $b_1 = 2.334733$; $b_2 =$

5  $0.250621$; $c_1 = 3.330657$; $c_2 = 1.681534$; $\gamma_1 = 1.7810$.

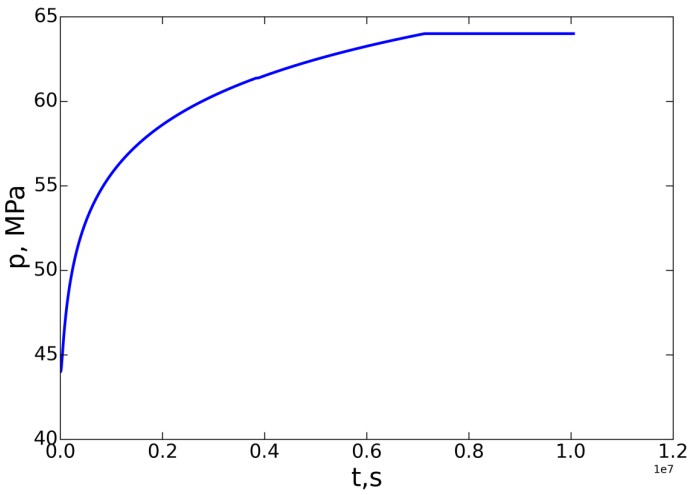

**Figure 3: Pore pressure change at the boundary between the first block and substrate.**

### 3. Results

To study the influence of the number of blocks in the multi-degree-of-freedom spring-block system on characteristics of

10  simulated seismicity (the total number of events, maximum and cumulative seismic moments) for "natural" and "induced"

cases, the calculations for 2, 3, 4, 5, 10, 15, 20 and 25 blocks were made for the same motion durations - one million

seconds. The time restriction was related with computational complexity of simulation for 25 blocks. During this time, the

pressure in set 2 changed significantly (near 11 MPa). The number of events, the maximum seismic moment of one event

and the cumulative seismic moment of all events and all blocks are shown in Figs. 4-15. The calculations were made for

15  different ratios of stiffness of the springs between the blocks $k_l$ to stiffness of links between the driver and the blocks $k_s$.



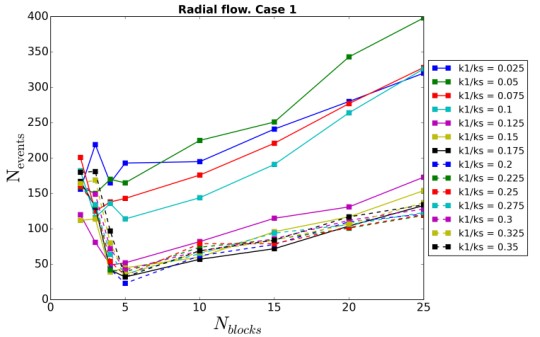

**Figure 4: Cumulitive number of events vs. number of blocks.**

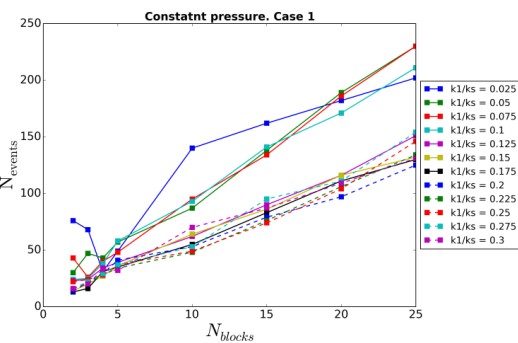

**Figure 5: Cumulitive number of events vs. number of blocks.**

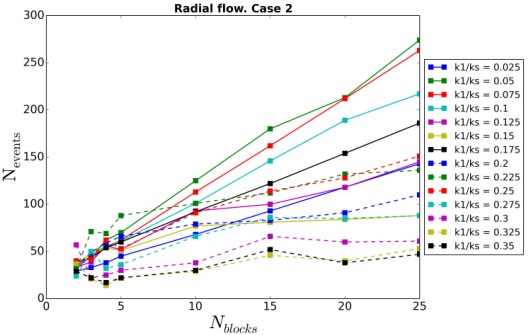

**Figure 6: Cumulitive number of events vs. number of blocks.**

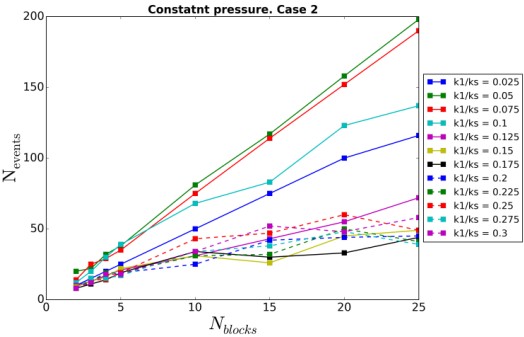

**Figure 7: Cumulitive number of events vs. number of blocks.**

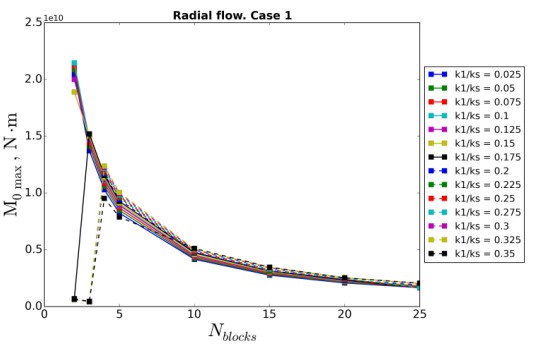

**Figure 8: Maximum seismic moment of event vs. number of blocks.**

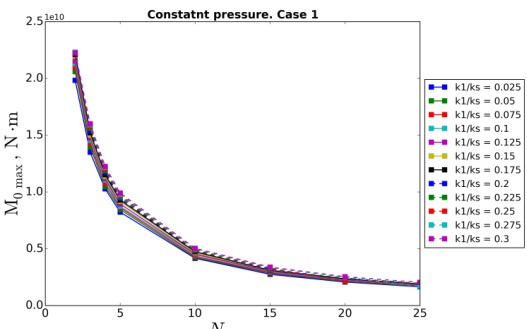

**Figure 9: Maximum seismic moment of event vs. number of blocks.**





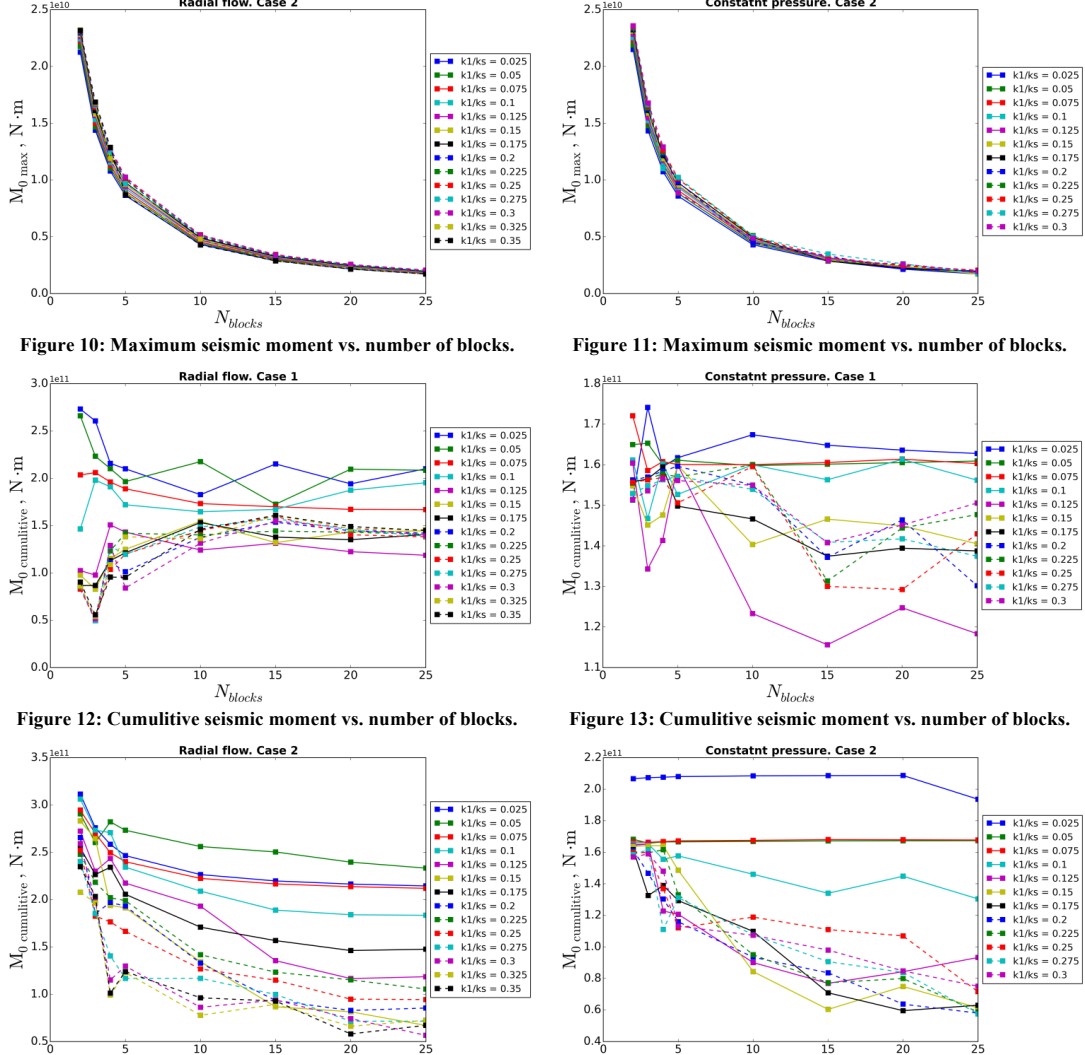

**Figure 10: Maximum seismic moment vs. number of blocks.**

**Figure 11: Maximum seismic moment vs. number of blocks.**

**Figure 12: Cumulitive seismic moment vs. number of blocks.**

**Figure 13: Cumulitive seismic moment vs. number of blocks.**

**Figure 14: Cumulitive seismic moment vs. number of blocks.**

**Figure 15: Cumulitive seismic moment vs. number of blocks.**

It can be seen, that if the pore pressure did not change (set 1, Fig. 5, Fig. 7, Fig. 9, Fig. 11), the number of events is growing almost linearly with the increase of the number of blocks for all values of stiffness of springs between the blocks in both Cases (1 and 2); the maximum seismic moment of the events is decreasing with the increase of the numbers of blocks.


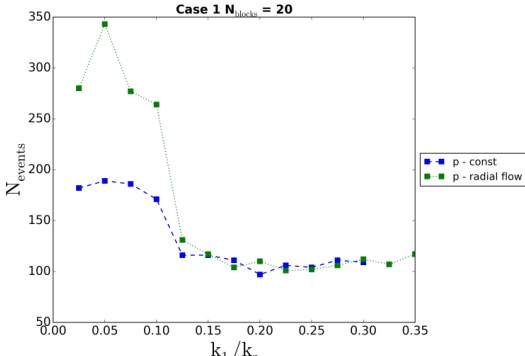

Figure 16: Dependence of cumulitive number of events on stiffnes of interblock link.

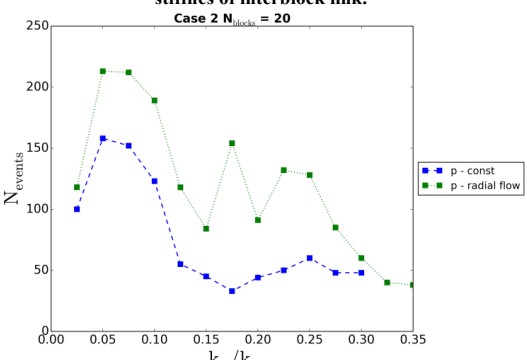

Figure 17: Dependence of cumulitive number of events on stiffnes of interblock link.

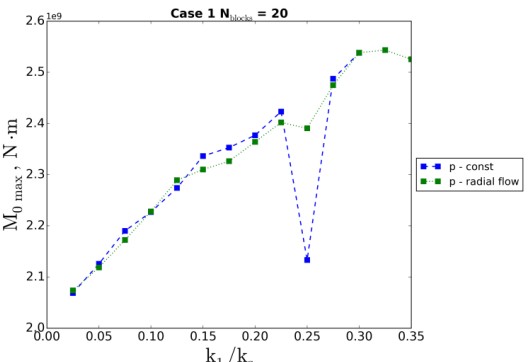

Figure 18: Dependence of maximum seismic moment of the event on stiffnes of interblock link.

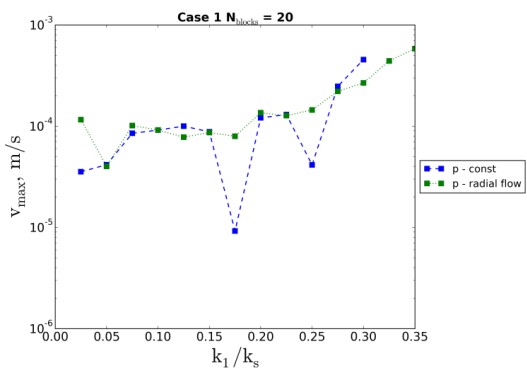

Figure 20: Dependence of maximum velocity on stiffnes of the interblock link.

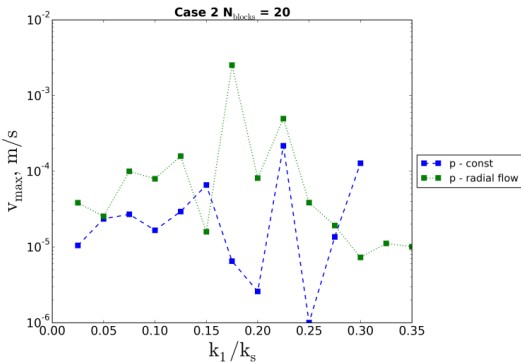

Figure 21: Dependence of maximum velocity on stiffnes of the interblock link.

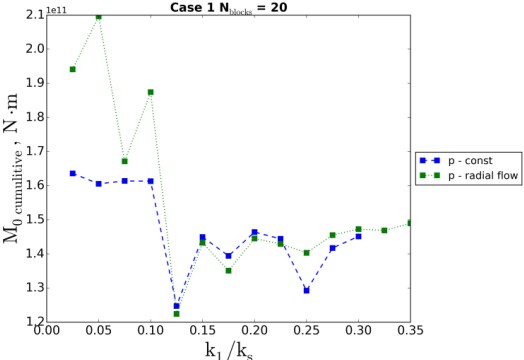

Figure 22: Dependence of cumulitive seismic moment for whole system on stiffnes of interblock link.

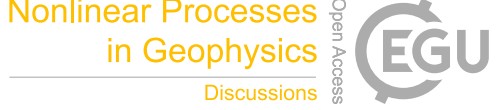


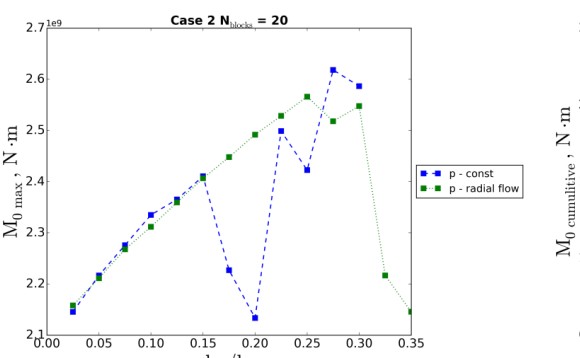
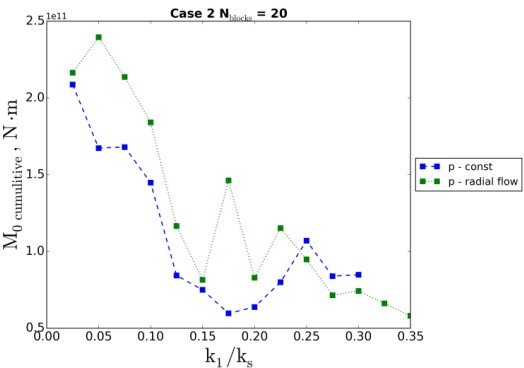

**Figure 19: Dependence of maximum seismic moment of the event on stiffnes of interblock link.**  **Figure 23: Dependence of cumulitive seismic moment for whole system on stiffnes of interblock link.**

However, for small values of $k_1$ (equal to $0.025k_s$) the total seismic moment doesn't depend on the number of blocks for both cases. In Case 1 and $k_1 > 0.1k_s$ for $N_{blocks} \leq 10$ the cumulative seismic moment slightly decreases, for $N_{blocks} > 10$ it almost does not change. In Case 2 the cumulative seismic moment decreases with increase of the number of the blocks. For set 2 with the pore pressure increase (Fig. 4, Fug. 6, Fig. 8, Fig. 10), the dependence is more complicated: in Case 1 for $k_1 \leq$

$0.1k_s$, the number of events also grows linearly with the increase of the number of blocks, but for $0.1k_s < k_1 \leq 0.35k_s$ the number of events decreases with the increase of the number of the blocks up to 5, and only then it starts to increase linearly. The maximum seismic moment decreases in both cases; deviation of one point for Set 2 ("induced" seismicity simulation) from the main trend is caused by insufficient calculating time. The total seismic moment almost does not change in Case 1, and gradually decreases in Case 2.

Now, let us consider the change of the behavior of the system consisted of 20 blocks with the change of stiffness of the link between the blocks $k_1$. In Case 1 (both Sets, Figs. 16, 18, 20) the total number of events initially decreases with the increase of $k_1$ and then stabilizes at value around 100, while the maximum seismic moment and the maximum block velocity increase almost monotonically. These results can be explained by the following. The Case 1 corresponds to chaotic behavior of the one-block system; the characteristic feature of that behavior is the quick changes in the block velocity. If there are many

blocks, the interaction of one block with its neighbors prevents significant increase of the block velocity. At low values of $k_1$ every neighboring block reacts with time lag to movement of the first block, and all blocks are moving asynchronously and disturb each other. The same effect causes large number of events. With increase of $k_1$, the first block perturbation transfers faster to other blocks; the system starts moving more synchronously, which leads to an increase of the block velocities and of the event seismic moments. At the same time, the total number of perturbations experienced by each block decreases, which

leads to the decrease of the number of events. All these features are illustrated in Figs. 24, 25. For convenience, we consider a short period of time and truncate the maximum value of velocity.




The Case 2 is characterized by slower changes of velocity with time than the Case 1 (Fig. 26). That is why there is no clear dependence of number of events and block maximum velocity on the interblock link stiffness. Such a behavior becomes more evident with decrease of parameter $A$.

Our model demonstrates that influence of the interblock link stiffness on behavior of studied systems is very strong. By
changing the stiffness, we may get periodic or chaotic motion of the system, occurrence of the first strong seismic event close to the injection start or after a relatively long time (compare Fig. 27 and 28); furthermore, the main seismic activity may occur at the injection start, when the pressure gradient is the highest, or in the post-injection phase. In Figs. 27-30 the seismic activity variations in the form of number of events per 10 days (left vertical axis) and the ratio of cumulative seismic moment of events to average cumulative seismic moment per 10 days (right vertical axis) are shown for both "natural" (set
1) and "induced" (set 2) seismicity. The "natural" seismic activity variations have almost the same amplitudes during all considered time interval, while the "induced" seismic activity variations depend on interblock link stiffness: in the case of small stiffness the amplitude of seismic activity during injection is almost the same as in the post-injection period (Fig. 29). When the stiffness becomes higher, the seismicity during injection becomes twice greater than post-injection activity (Fig. 30); further increase of the interblock link stiffness leads to significant increase of the post-injection activity (Figs. 31, 32).

The recurrence maps of seismic event sequences are shown in Figs. 33, 34. It could be seen that for $k_l/k_s$=0.25 the time intervals between two events converge to several points both for "induced" and "natural" seismicity (only post-injection seismic activity is considered). For $k_l/k_s$=0.3, the "natural" seismicity shows periodic variations, while "induced" seismicity has more complicated chaotic behavior. For other values of $k_l$ both "induced" and "natural" seismicity shows chaotic variations.


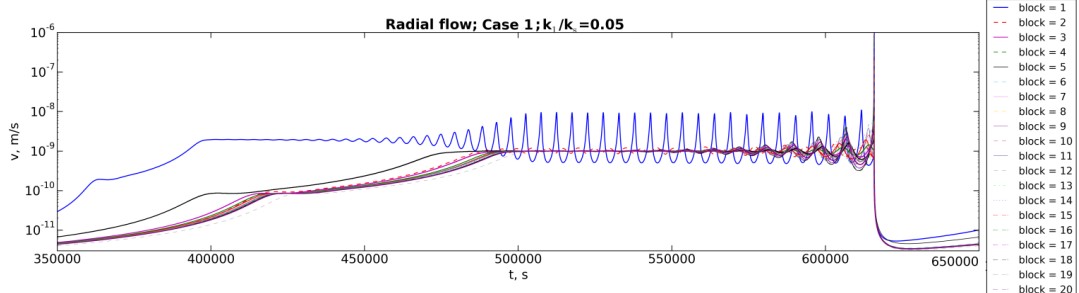

**Figure 24: Block velocity variations in time for system consisted of 20 blocks in Case 1.**





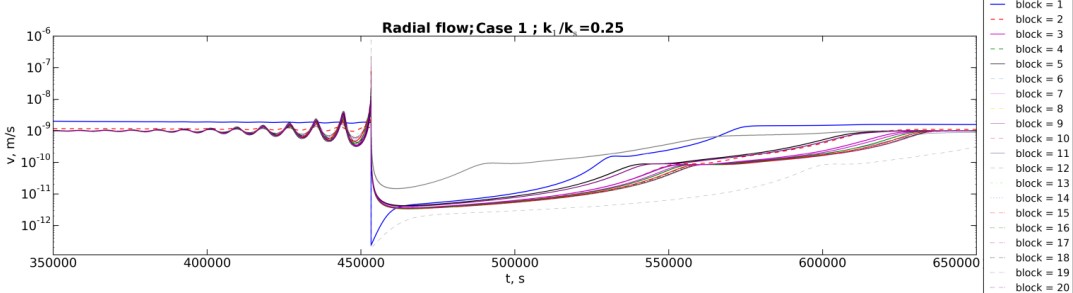

**Figure 25: Block velocity variations in time for system consisted of 20 blocks in Case 1.**

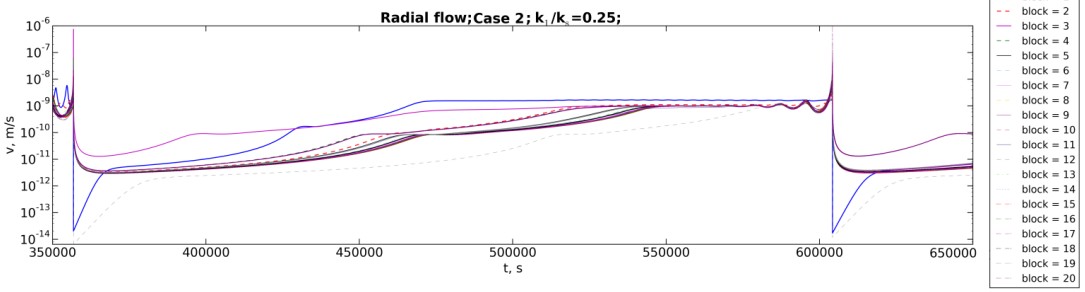

**Figure 26: Block velocity variations in time for system consisted of 20 blocks in Case 2**

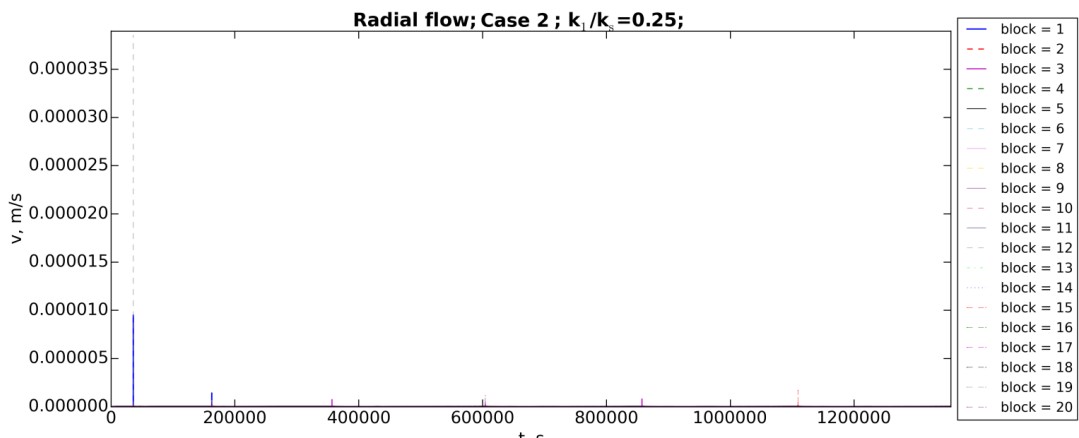

**Figure 27: Seismic event occurrences in time for system consisted of 20 blocks in Case 2.**





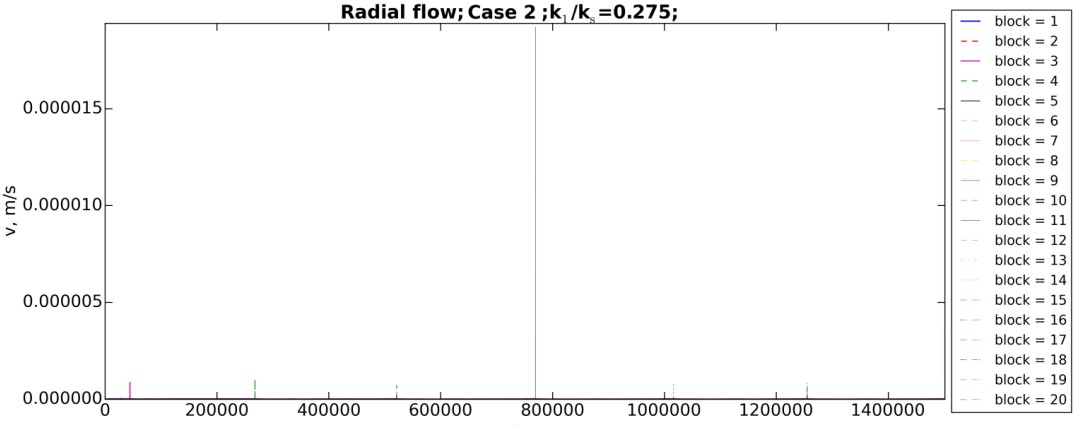

Figure 28: Seismic event occurrences in time for system consisted of 20 blocks in Case 2.

## 4. Discussion and conclusions

The problem of the influence of fluid injection on tectonic fault sliding and generation of seismic events was studied by
numerical calculations of the peculiarities of motions of a system of blocks (consisted from up to 25 blocks) elastically
connected with each other and connected by elastic springs to a constant-velocity moving driver (multi-degree-of-freedom
spring-block model). The rate-and-state friction model with two-parametric friction law was adopted for description of
friction between the blocks and the substrate. Initially the block system was in steady-sliding state, then its state was
disturbed by the pore pressure increase. Influences of the model parameters (number of the blocks, the spring stiffness,
velocity weakening parameter) on the process of the model seismicity variations were considered.

It was shown that considered spring-block system could exhibit different types of motion with different patterns. The motion
could be periodic or chaotic; the magnitude of the seismic events depends on fragmentation of the fault system (the number
of blocks in considered model) and may have different values. The analysis shows that the stiffness of link between the
blocks affect significantly the behaviour of the model and resulting seismicity, so the main seismic activity could appear
directly after the start of fluid injection or in the post-injection phase. Such influence of injection on seismicity could be
observed in the real cases. However, the parameters in the rate-and-state model are known only from laboratory experiments,
and it is hard to believe that one should use the same values to describe the real scale phenomena. Yet our study showed, that
it is possible to select more suitable parameters that will allow one to match results of calculations and data of real
observations. It can be concluded, that considered model has the potential to be used for the estimations of the possible fluid-
induced seismicity activity variations.



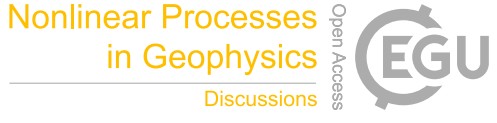
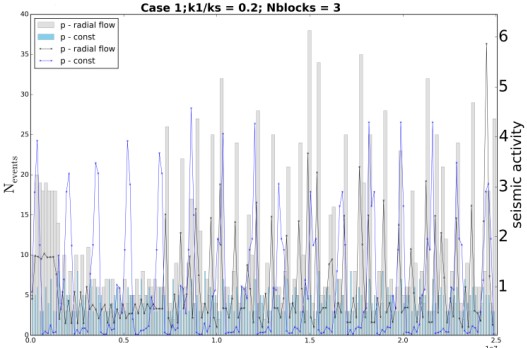

**Figure 29: Time variation of seismic activity.**

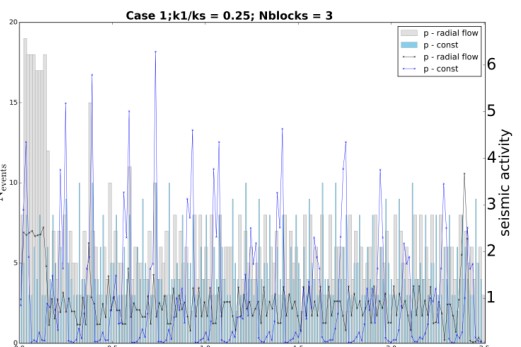

**Figure 30: Time variation of seismic activity.**

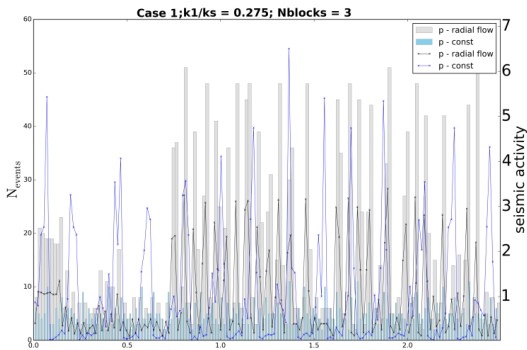

**Figure 31: Time variation of seismic activity.**

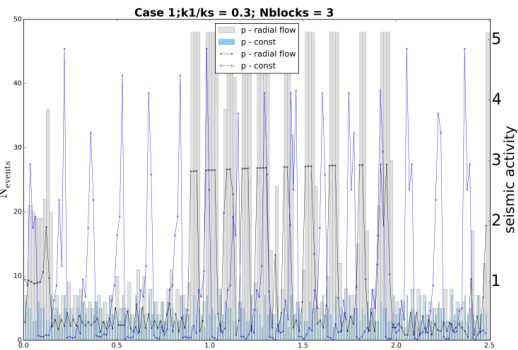

**Figure 32: Time variation of seismic activity.**

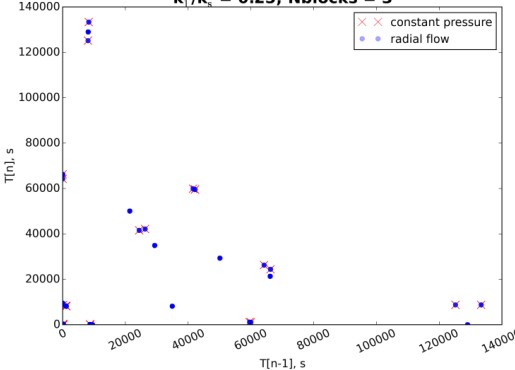

**Figure 33: Iteration map of recurrence intervals of seismic events, $T_n$ denotes the time interval between $n_{th}$ and $(n+1)_{th}$ events. The map includes events occurred at time $t \geq 8 \cdot 10^6\ s$.**

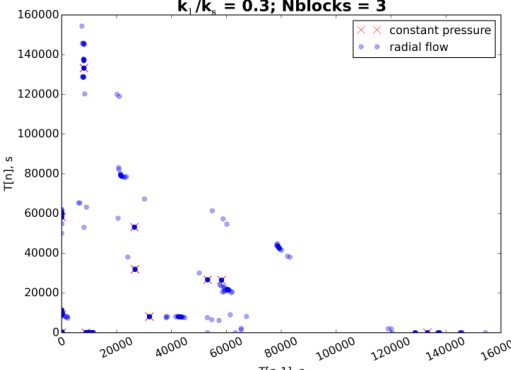

**Figure 34: Iteration map of recurrence intervals of seismic events, $T_n$ denotes the time interval between $n_{th}$ and $(n+1)_{th}$ events. The map includes events occurred at time $t \geq 8 \cdot 10^6\ s$.**



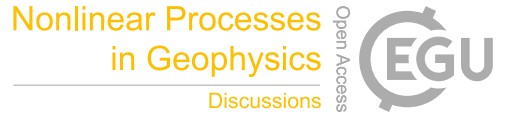

**Acknowledgements**

The financial support of the Russian Foundation for Basic Researches (project # 16-05-00869) is acknowledged.

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
