# Peer review of "Non-linear effects of pore pressure increase on seismic event generation in multi-degree-of-freedom rate-and-state model of tectonic fault sliding."

_Nonlinear Processes in Geophysics, 2016_

## Referee Comment (RC1) · Anonymous Referee #1 · 1 Jan 2017

The paper presents simulations of the fault movement caused by the spatial friction variations modelling fluid injection. The paper is interesting and I would recommend it for publication subject to minor revision to address the following:

1. Figures 27, 28 are almost empty, that is small variations in velocities are not seen. Possibly, using log scale in the velocity axis could help. 2. Captions to Figs. 16-22, should be dependence of …. of rather than …on. 3. Discussion, line 11, "considered spring-block system ", should be "the considered spring-block system"; line 14, "affects" rather than "affect"

---

## Author Comment (AC1) · 16 Jan 2017

We would like to thank very much the anonymous reviewer for attention to our paper and for thorough reading of the paper. All the suggestions to improve the text and the figures were accepted and the paper was corrected in accordance with the reviewer remarks.

Sergey Turuntaev Vasily Riga

Please also note the supplement to this comment:

http://www.nonlin-processes-geophys-discuss.net/npg-2016-66/npg-2016-66-AC1-supplement.pdf

[Figure]

**Supplement:**

[revised manuscript text omitted]

**Acknowledgements**

The financial support of the Russian Foundation for Basic Researches (project # 16-05-00869) is acknowledged.

---

## Referee Comment (RC2) · Anonymous Referee #2 · 7 Mar 2017

The authors have presented interesting results of a numerical study on nonlinear effects of pore pressure increase on seismic event generation in multi-degree-of-freedom rate-and-state model of tectonic fault sliding. The paper is worth to be published after taking into account the items that follow.

1. English needs a thorough review. It is the author's responsibility to revise the English in the entire manuscript to a much higher standard. 2. Expressions relating the friction forces in the Eq. (1) and shear stress are absent, of course, it is evident, but it is necessary to note. 3. Please emphasize the differences with the literatures

Häring et al., 2008, Dinske, 2010.

Please also note the supplement to this comment:
http://www.nonlin-processes-geophys-discuss.net/npg-2016-66/npg-2016-66-RC2-supplement.pdf

**Supplement:**

**Review of NPG-2016-66, by Sergey B. Turuntaev and Vasily Y. Riga,**
**"Non-linear effects of pore pressure increase on seismic event generation in multi-degree-of-freedom rate-and-state model of tectonic fault sliding"**

The authors have presented interesting results of a numerical study on nonlinear effects of pore pressure increase on seismic event generation in multi-degree-of-freedom rate-and-state model of tectonic fault sliding. The paper is worth to be published after taking into account the items that follow.

1. English needs a thorough review. It is the author's responsibility to revise the English in the entire manuscript to a much higher standard.

2. Expressions relating the $F_{fr}i$ in the Eq. (1) and shear stress are absent, of course, it is evident, but it is necessary to note.

3. Please emphasize the differences with the literatures Häring et al., 2008, Dinske, 2010.

---

## Author Comment (AC2) · 28 Mar 2017

We would like to thank very much anonymous referee for the attention to our paper and for thorough reading of the paper. All the suggestions to improve the text were accepted and the paper was corrected in accordance with the referee remarks.

1. English needs a thorough review. It is the author's responsibility to revise the English in the entire manuscript to a much higher standard. We sent the paper to English speaking person, all suggested corrections were adopted.

2. Expressions relating the friction forces in the Eq. (1) and shear stress are absent, of

course, it is evident, but it is necessary to note. We added expression related friction force and shear stress.

3. Please emphasize the differences with the literatures Häring et al., 2008, Dinske, 2010. We relocated references to the two literatures to emphasize that the value of hydraulic diffusion coefficient was taken from Dinske, 2010, other values were chosen based on Häring et al., 2008.

Sergey Turuntaev Vasily Riga

---

## Author Response (AR1)

We would like to thank very much anonymous referees for attention to our paper and for thorough reading of the paper. All the suggestions to improve the text and the figures were accepted and the paper was corrected in accordance with the referee remarks.

Anonymous Referee #1

1. Figures 27, 28 are almost empty, that is small variations in velocities are not seen. Possibly, using log scale in the velocity axis could help.

We changed the Figures 27, 28 using log scale.

2. Captions to Figs. 16-22, should be dependence of . . .. of rather than . . .on.

Captions to Figs. 16-22 were re-written.

3. Discussion, line 11, "considered spring-block system ", should be "the considered spring-block system"; line 14, "affects" rather than "affect"

Suggested corrections were adopted.

Anonymous Referee #2

1. English needs a thorough review. It is the author's responsibility to revise the English in the entire manuscript to a much higher standard.

We sent the paper to English speaking person, all suggested corrections were adopted.

2. Expressions relating the friction forces in the Eq. (1) and shear stress are absent, of course, it is evident, but it is necessary to note.

We added expression related friction force and shear stress.

3. Please emphasize the differences with the literatures Häring et al., 2008, Dinske, 2010.

We relocated references to the two literatures to emphasize that the value of hydraulic diffusion coefficient was taken from Dinske, 2010, other values were chosen based on Häring et al., 2008.

Sergey Turuntaev

Vasily Riga

[revised manuscript text omitted]

**Acknowledgements**

The financial support of the Russian Foundation for Basic Researches (project # 16-05-00869) is acknowledged.